# Scene-Clipping Long Video For Better Understanding

**Ziyu Zhao**
2024213688
School of Software

**Jin Wang**
2024213687
School of Software

**Jinsong Xiao**
2024210747
Department of Electrical Engineering

## 1   Introduction

Recent advances in large-scale video-language models, such as GPT-4o and Gemini-1.5-Pro, have showcased their remarkable ability to understand long video content, due to their support for long context length. These models exhibit impressive potential for deep comprehension of video content, particularly in tasks that require real-time analysis by processing ongoing sequences and retrieving information from long-term memory. However, training such foundational models at this scale remains out of reach for most academic researchers because of the immense computational resources needed to handle the high-dimensional complexity of long-video data. Many current open-source large multimodal models concatenate the query embeddings of each frame along the time axis and input them into the LLM. Although this approach has shown promising results, particularly with short videos, it faces significant challenges when applied to long videos. Consequently, this design becomes impractical for longer videos, as the inherent context length limitations of LLMs and the high GPU memory consumption severely restrict the number of frames that can be processed. For instance, LLaMA has a context length limitation of 2048 tokens, while large multimodal models like LLaVA(1) and BLIP-2(2) can only process 256 and 32 tokens per image respectively.

To address these challenges, there has been a growing interest in developing efficient Video-LLMs that can efficiently process long video sequences despite restricted context length. VideoChat(3), Video-LLaVA(4) and Video-Llama(5) convert a fixed number of sampled frames into a small number of embeddings, regardless of the video's duration, resulting in inadequate information for effectively representing long videos. Both MA-LMM(6) and MovieChat(7) utilize memory-augmented mechanisms to extend the context window for processing long-form video content, allowing them to retain and reference information over extended time periods. However, this memory-averaging approach can lead to a gradual reduction in the richness of the retained information, as it tends to compress and dilute details over time. This can result in an uneven representation, where earlier frames or key moments lose significance, making it challenging to maintain a balanced and detailed understanding of the entire video. TimeChat(8) and LVCHAT(9) group the original video frames and then apply specific aggregation techniques to reduce the number of tokens, achieving more efficient compression. However, the group size must be predetermined and remains fixed, limiting the model's ability to adapt to the unique characteristics of each video. Additionally, the video content within the same group may vary significantly, which can lead to a substantial loss in representational quality after aggregation, hindering the model's ability to accurately capture critical details. Chat-UniVi(10) and VideoLLaMB(11) reduce information loss during aggregation by segmenting the video into distinct segments based on scene changes, helping to preserve semantic coherence. However, these methods still require predefined segmentation ratios or a fixed number of segments, limiting their flexibility in adapting to different video content. Moreover, Chat-UniVi's DPC-KNN-based scene segmentation algorithm can disrupt the original temporal sequence, potentially affecting the natural flow of events within the video.

In light of these challenges, we propose our scene-clipping long video LLM, a novel approach that aggregates spatiotemporal context across extended temporal horizons. To address the limitations of the aforementioned scene segmentation algorithms, we propose a dynamic scene-clipping algorithm that partitions the original video into clips based on the specific scene distribution, eliminating the need to pre-specify the number of clips. This approach ensures semantic consistency within each

Preprint. Under review.

clip. Subsequently, we utilize video-Qformer to extract features from each clip while incorporating temporal encoding information, enabling the LLM to comprehensively understand the spatio-temporal content of the long video.

## 2 Related Work

### 2.1 Video-LLMs

Recent Video-LLMs have made strides in improving the understanding of temporal dynamics in video content. For instance, Video-Llama(5) enhances the BLIP-2 architecture by introducing an additional video-querying transformer to explicitly model temporal relationships. Similarly, Video-ChatGPT(12), built on LLaVA, employs a simple average pooling of frame-level features across spatial and temporal dimensions to generate a unified video-level representation. Meanwhile, VideoChat(3) employs perception models to generate action and object annotations, which are then processed by LLMs for higher-level reasoning. Building on these advances, VideoChat2(13) introduced a multi-stage bootstrapping technique focused on modality alignment and instruction tuning, allowing the collection of high-quality video data for fine-tuning instruction-driven tasks. Video-LLaVA(4) enhances modality integration by using a pre-aligned encoder adaptable to both images and videos which enables shared projections and synergistic training across image and video tasks. Although these models represent significant advances, they are predominantly designed for short videos. Longer videos present considerable challenges due to the inherent limitations of LLM context length and the high memory demands on GPUs. These factors restrict the ability of current models to scale effectively for long-term video understanding.

### 2.2 Long-term Video-LLMs

Long-term Video-LLMs aim to capture extended patterns in videos that typically exceed 30 seconds in duration. Long videos pose challenges due to high computational complexity and memory demands, prompting long-term video LLMs to adopt advanced temporal modeling techniques for improved efficiency. MovieChat(7) introduced a novel memory-based mechanism that strategically merges similar frames to reduce computational load and memory usage. Chat-UniVi(10) proposed a unified approach to processing images and videos by dynamically merging similar spatial and temporal tokens to improve efficiency. LLaMA-VID(14) condensed video representations by representing each frame with only two tokens, separating context and content tokens for more efficient compression. For long video QA, Xu *et al.*(15) explore selectively using frames or clips from long videos using retrieval-based methods. This approach aims to focus on the most relevant video segments, improving efficiency and effectiveness in answering questions based on extended video content. TimeChat and LVCHAT group the original video frames and apply specific aggregation techniques to reduce the number of tokens, thus achieving more efficient compression.

## 3 Method

We propose a fine-tuning approach that leverages a frozen video LLM integrated with a Video-Qformer, pre-trained on short video clips, to adapt it for long video content. Given a video $V$ with $n$ frames, we first extract frames to obtain a complete sequence of frame representations $F = \{f_1, f_2, ..., f_n\}$ using the pre-trained image encoder. Next, we apply our entropy-based scene-clipping algorithm to frame embeddings $F$ to generate $k$ clips. The frame embeddings within each clip are then fed into the Video-Qformer to obtain clip embeddings $C = \{c_1, c_2, ..., c_k\}$, incorporating temporal position information, which are then concatenated to produce the final representation. This approach enables the LLM to comprehensively understand the spatiotemporal content of long videos.
**Datasets** Long videos data from VideoChat2(13) and ShareGPT4Video(16). Long video dataset ActivityNet Captions(17).
**Baselines** Video-Llama(5), VideoLLaMB(11), Chat-Univi(10), MovieChat(7) and Timechat(8).
**Benchmarks** MVBench(13) for short video understanding. EgoSchema(18) for long video QA. LongVideoBench(19) and MLVU(20) for multi-task long video understanding.

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
