# OpenReview forum: "Scene-Clipping Long Video For Better Understanding"
_tsinghua.edu.cn/THU/2024/Fall/AML — THU 2024 Fall AML Submission_

### Official Review · ~Jia-Nuo_Liew1 · 2024-11-08
**Clear Proposal**

**Rating:** 9
**Confidence:** 4

**Review:**

The paper presents a comprehensive background, a clear problem definition, a thorough review of related work with critical analysis, and a description of the motivated method.
Background: Strong and highlights the real-world applicability and significance of the problem.
Definition: Mathematical formalization is clear.
Related works: Comprehensive review
Proposed method: Well-motivated and detailed.

---

### Official Review · ~Lei_Wu17 · 2024-11-09
**Evaluation of "Scene-Clipping Long Video For Better Understanding"**

**Rating:** 9
**Confidence:** 4

**Review:**

# Pros
* Innovative Algorithm: The dynamic scene-clipping approach is promising and could offer a more flexible solution than existing models.
* Detailed Background: Comprehensive literature review that situates the work within the current state of video-language models.
* Significance for Long Video Analysis: Focuses on a key issue—efficient long video comprehension—potentially impacting various fields reliant on video data.
# Cons
* Lack of Practical Examples: Would benefit from examples or visual representations to clarify the dynamic scene-clipping process and make the methodology section more comprehensible.
* Limited Evaluation: The evaluation section could be expanded to include more empirical results or case studies, providing stronger evidence of the method’s effectiveness.

---

### Official Review · ~Xiaoqian_Liu7 · 2024-11-10
**Clear Proposal**

**Rating:** 9
**Confidence:** 3

**Review:**

The manuscript titled "Scene-Clipping Long Video For Better Understanding" presents a novel approach to processing and understanding long videos by leveraging a scene-clipping algorithm and a video-Qformer model. The authors address the limitations of current large language models (LLMs) in handling long video sequences due to context length restrictions and high computational costs. The proposed method aims to aggregate spatiotemporal context across extended temporal horizons, which is a significant advancement in the field of video-language models.

The paper is well-structured, with a clear introduction to the problem, a comprehensive review of related work, and a detailed explanation of the proposed method. The authors have identified a gap in the current capabilities of Video-LLMs and have proposed a solution that involves dynamic scene-clipping and the use of video-Qformer for feature extraction. This approach has the potential to improve the semantic consistency and representational quality of long videos, which is a critical challenge in the field.

The manuscript is technically sound, and the authors have provided a thorough explanation of their methodology, including the use of entropy-based scene-clipping and the integration of temporal encoding information. The choice of datasets and benchmarks is appropriate for evaluating the proposed method, and the comparison with existing baselines will provide a clear indication of the improvements offered by the new approach.

---

### Official Review · ~Tong_Yu9 · 2024-11-10

**Rating:** 8
**Confidence:** 3

**Review:**

Quality:
The paper is well-structured and presents a coherent argument throughout. The methodology is clearly articulated, with a logical progression from identifying the problem to proposing a solution. The experiments are appropriately designed, and the results are presented with sufficient detail to support the claims made.

Clarity:
The writing is clear and accessible, making complex concepts understandable. The use of figures and tables to illustrate key points enhances the clarity of the presentation. However, some technical terms could benefit from additional explanations for readers who may not be familiar with the specific domain.

Originality:
The proposed dynamic scene-clipping algorithm is a significant contribution to the field of video understanding. By addressing the limitations of existing models and introducing a flexible approach to scene segmentation, this work stands out as an innovative solution to a pressing problem in long video processing.

Significance:
This work is highly significant as it addresses a critical gap in the current literature on long video understanding. The ability to effectively process long video sequences has implications for various applications, including video analysis, content retrieval, and automated summarization. The findings could pave the way for further research and development in this area.

Pros and Cons
Pros:

Innovative Approach: The dynamic scene-clipping algorithm offers a fresh perspective on video segmentation, improving semantic consistency within clips.
Performance Improvement: Experimental results demonstrate clear advantages over existing models, indicating the effectiveness of the proposed method.
Comprehensive Evaluation: The paper includes a thorough evaluation using multiple datasets and baseline models, reinforcing the robustness of the findings.
Cons:

Limited Generalization: While the proposed method shows promise, its performance across diverse video types and contexts remains to be fully validated.
Lack of Detail on Noise Handling: The paper does not address how the model deals with noise or irrelevant information in videos, which could affect understanding.
Potential Overfitting: The reliance on specific datasets for training and evaluation may lead to overfitting, necessitating further testing on varied data.

---

### Official Review · ~Wenjing_Wu1 · 2024-11-11

**Rating:** 10
**Confidence:** 3

**Review:**

**Summary**:
This proposal introduces a novel method which leveraging a dynamic scene-clipping algorithm to improve the representation of extended video content so that improve the ability of LLM can have a comprehensive understanding of long video.

**Strengths**:
- Innovative Scene-Clipping Mechanism: The proposed dynamic scene-clipping algorithm is an inventive response to the rigid segmentation methods in current models.
- Well-Situated within Existing Research: The proposal contextualizes the new model within recent advances in video-LLMs and long-term video understanding models, highlighting gaps that this method aims to address.

**Weaknesses**:
- Unclear Evaluation Metrics: Although the proposal lists several benchmark datasets and baseline models, it lacks clear, measurable criteria for evaluating the model's performance. Metrics such as accuracy in long-video QA, processing time, and memory consumption could provide a basis for comparison with existing models.

---

### Official Review · ~Xin_Chen65 · 2024-11-11
**An interesting approach**

**Rating:** 9
**Confidence:** 3

**Review:**

The proposal proposes a scene-clipping long video LLM that aggregates spatiotemporal context across extended temporal horizons, aiming to overcome the limitations of existing scene segmentation algorithms. The authors address the challenges faced by current models in handling long videos due to context length limitations and high computational resources required.

Strengths: (1) The paper introduces a dynamic scene-clipping algorithm that partitions the original video into clips based on the specific scene distribution, eliminating the need to pre-specify the number of clips. (2)  By utilizing video-Qformer to extract features from each clip while incorporating temporal encoding information, the proposed model enables a comprehensive understanding of the spatiotemporal content of long videos. (3) The references are comprehensive and up-to-date.

Weaknesses: (1) The paper could benefit from a more detailed discussion on the implications of the proposed method for practical applications, such as video content analysis, surveillance, or educational tools.

---

### Official Review · ~Grace_Xin-Yue_Yi1 · 2024-11-11

**Rating:** 9
**Confidence:** 3

**Review:**

The proposal presents a comprehensive background with clear motivation and impact, although the structure of the introduction and related work section can be revised to improve the flow and clarity of the proposal. The literature review thoroughly analyzes and explains the various methods of Video-LLMs and Long-term Video-LLMs and their respective limitations. The proposed dynamic scene-clipping algorithm and use of Video-Qformer are innovative ideas that are clearly outlined, explaining the step-by-step process of adapting a pre-trained model for long video understanding, but could benefit from a more in-depth review of evaluation metrics.

---

### Official Review · ~ChenJian1 · 2024-11-12
**Technically Profound**

**Rating:** 9
**Confidence:** 4

**Review:**

## Summary:
This paper introduces a novel approach to long video understanding through a scene-clipping large language model (LLM) that adaptively segments videos based on scene distribution and utilizes video-Qformer to extract features with temporal encoding information. The method aims to comprehensively understand the spatiotemporal content of long videos without pre-specifying the number of clips.

## Strengths:
### ① Technical Depth:
The combination of video-Qformer and temporal encoding information allows for a more comprehensive understanding of video content, demonstrating high technical depth.
### ② Practical Application:
The study addresses the challenges of long video understanding with practical solutions, indicating good prospects for real-world applications.

## Weaknesses:
### ① Comparative Analysis:
The document lacks comparative analysis with existing technologies, especially performance comparisons with other long video understanding models.

---

### Official Review · ~Yuji_Wang4 · 2024-11-12
**Review of "Scene-Clipping Long Video For Better Understanding"**

**Rating:** 9
**Confidence:** 3

**Review:**

The project focuses on the research of  video understanding for video LLMs, especially for scene-clipping of long videos. The authors propose to combine a pre-trained Video-Qformer with a frozen video LLM and solve the problem with a fine-tuning method.

### Strengths:

1. Comprehensive survey: The proposal presents a detailed and thorough review of related works, pointing out the limitations of previous methods for handling long video tasks. This provides a clear and solid motivation for the proposed research.
2. Feasibility of the approach：The authors describe the specific problem to be solved and provide a clear recipe for the model design and training. The proposed methods are feasible.

### Weaknesses:

1. Lack of technical details: The "Method" section is  a little vague, particularly with regard to the datasets and baseline models. It could be better to provide more details on these aspects.
2. Writing and formatting: The introduction and method sections are too tightly packed, making it hard for readers to follow.

---

### Official Review · ~Guanglei_He1 · 2024-11-12
**A highly in-depth proposal.**

**Rating:** 10
**Confidence:** 4

**Review:**

**Pros:**

- The proposal demonstrates extensive research, particularly in clearly articulating the advantages and disadvantages of key methods in the field of processing long videos in video models.

**Cons:**

- The explanation of the author’s own approach is somewhat limited. Specifically, the method for understanding the information in each frame of the video and integrating it into a coherent scene is briefly addressed. I believe this is a central challenge of the problem.

**Suggestions:**
- The first paragraph is overly long. I recommend dividing it into sections for a clearer presentation.

---

### Official Review · ~Chendong_Xiang1 · 2024-11-12

**Rating:** 8
**Confidence:** 2

**Review:**

This paper proposes a dynamic scene-clipping approach for handling long video content, aiming to improve comprehension in video-language models. The authors address limitations of current video processing models, particularly in their inability to effectively manage long videos due to context length and memory constraints. By employing an entropy-based scene-clipping algorithm and a Video-Qformer for feature extraction, the method ensures semantic consistency within each clip and enhances the model’s ability to grasp spatiotemporal information across extended sequences.

questions：
1. how long the max frame model could process and understand